

**Rain concentration and sheltering effect of solar panels on cultivated plots**
Yassin Elamri[1,2], Bruno Cheviron[3], Annabelle Mange[4], Cyril Dejean[5], François Liron[6], Gilles Belaud[7]
[1] IRSTEA/UMR G-Eau, 361 rue Jean-François Breton 34136 Montpellier (FRANCE), yassin.elamri@irstea.fr
[2] Sun'R sas, 41 quai Fulchiron 69005 Lyon (FRANCE), yassin.elamri@sunr.fr
[3] IRSTEA/UMR G-Eau, 361 rue Jean-François Breton  34136 Montpellier (FRANCE), bruno.cheviron@irstea.fr
[4] IRSTEA/UMR G-Eau, 361 rue Jean-François Breton  34136 Montpellier (FRANCE), annabelle.mange@irstea.fr
[5] IRSTEA/UMR G-Eau, 361 rue Jean-François Breton 34136 Montpellier (FRANCE), cyril.dejean@irstea.fr
[6] IRSTEA/UMR G-Eau, 361 rue Jean-François Breton 34136 Montpellier (FRANCE), francois.liron@irstea.fr
[7] Montpellier SupAgro/UMR G-Eau, 2 place Pierre Viala 34060 Montpellier (FRANCE), belaud@supagro.inra.fr





**Abstract**
Agrivoltaism is the association of agricultural and photovoltaic energy production on the same land
area, coping with the increasing pressure on land use and water resources while delivering a clean
and renewable energy. However the solar panels located above the cultivated plots also have a
seemingly unexplored yet effect on rain redistribution, sheltering large parts of the plot but
redirecting concentrated fluxes on a few locations. The spatial heterogeneity in water amounts
observed on the ground is high in the general case ; its dynamical patterns are directly attributable to
the mobile panels through their geometrical characteristics (dimensions, height, coverage
percentage) and the strategies selected to rotate them around their support tube. A coefficient of
variation is used to measure this spatial heterogeneity and to compare it with the coefficient of
uniformity that classically describes the efficiency of irrigation systems. A rain redistribution model
(AVrain) was derived from literature elements and theoretical grounds then validated from
experiments in both field and controlled conditions. AVrain simulates the effective rain amounts on
the plot from a few forcing data (rainfall, wind velocity and direction) thus allows real-time strategies
that consist in operating the panels so as to limit rain interception mainly responsible for the spatial
heterogeneities. Such avoidance strategies resulted in a sharp decrease of the coefficient of
variation, e.g. 0.22 against 2.13 for panels held flat during one of the monitored rain events, that is a
fairly good uniformity score for irrigation specialists. Finally, the water amounts predicted by AVrain
were used as inputs to HYDRUS-2D for a brief exploratory study on the impact of the presence of
solar panels on rain redistribution at shallow depths within soils : similar, more diffuse patterns were
simulated and coherent with field measurements.
**Copyright statement**
Data collection and model development were performed in the frame of the Sun'Agri2B project that
links the Sun'R SAS society with Irstea, SupAgro Montpellier and other academic or non-academic
partners. The copyright on all experimental and theoretical results presented here is governed by the
consortium agreement of the Sun'Agri2B project.



## 1. Introduction

The current climate change context induced by the production and consumption of highly polluting fossil energies, responsible for the greenhouse effect, has in turn triggered the development of clean and renewable energies with special interest for photovoltaic systems (IPCC, 2014). The recent times have seen a clear increase of land coverage by solar panels disposed on roofs, used for parking shadehouses or organized in solar farms (IPCC, 2011). In the last years, solar panels were installed above cultivated plots in France (Marrou, 2012), in Japan (Movellan, 2013), in India (Harinarayana and Vasavi, 2014), in the USA (Ravi et al., 2014) and in Germany (Osborne, 2016) so as not to create competition between different land uses (Dinesh and Pearce 2016). These innovative devices termed "agrivoltaic" by Dupraz et al. (2011) allow maintaining the agricultural yield under certain conditions (Marrou et al., 2013b; Marrou et al., 2013c), together with water savings (Marrou et al., 2013a) which results in the expected higher values of the dedicated "land use efficiency" indicator (Marrou 2012)

Besides blocking and converting a part of the incoming solar radiation, the implementation of solar panels in natural settings has a series of direct or indirect effects on several terms of the hydrological budget, in the equipped plots (Cook and McCuen 2013; Barnard et al. 2017). Although far less studied, these on-site or off-site hydrological consequences should be addressed and modeled for site preservation purposes in the general case and also because they are very likely to constrain the optimal irrigation and local site management strategies, on the cultivated plots. For example, Diermanse (1999) showed that a correct simulation of runoff could often be achieved at the watershed scale from spatially-averaged rainfall values, although clearly better results may be expected when explicitly accounting for the subscale spatial patterns of rain distribution (Faurès et al., 1995; Tang et al., 2007; Emmanuel et al., 2015). At the plot scale, rain interception and redistribution by the crops (Levia and Germer, 2015; Yuan et al., 2017) is already known to cause strong spatial heterogeneities (through stemflow, throughfall or improved water storage capabilities) thus to raise multiple questions on soil microbiology, non-point source pollution and irrigation piloting (Lamm and Manges, 2000; Martello et al., 2015). The presence of solar panels will provide similar, additional issues, close to these experienced in agroforestry when the vegetative cover is of various heights and nature, with a direct impact on the spatiotemporal patterns of rain redistribution (Jackson, 2000). More into details and more specifically, the interception of rain by the impervious surface of the solar panels produces an "umbrella effect" that delineates a sheltered area. By contrast, its contour receives the collected fluxes, whose intensity or amounts may locally exceed these of the control conditions, depending on the dimensions, height and tilting angle of the panels



as well as on wind velocity and direction. Cook and McCuen (2013) stated that one benefit of grass
growing was to damp or suppress any specific effect of solar panels on runoff at the plot scale. This
also constitutes valuable preventive measure against erosion issues arising from concentrated flows
in micro-gullies (Knapen et al., 2007; Gumiere et al., 2009) or attributable to the direct mechanical
effects of droplet impacts, known as splash erosion (Nearing and Bradford, 1985; Josserand and
Zaleski, 2003).

Agricultural soils should preferentially not be left bare under solar panel structures, because of
increased risks of runoff and erosion but these are only the most severe particular cases among the
diverse rain redistribution effects investigated in the present paper. These are possibly described
from geometrical arguments for an intuitive overview, suggesting three categories of zones on the
ground, in the agrivoltaic plots, (i) the non-impacted zones between panels that receive the same
rain amounts as the control site, (ii) the sheltered zones located right under the panels that receive
far less rainfall than in the control conditions and (iii) the border zones located where panels
discharge the collected rain amounts.

In most cultivated plots, the spatial heterogeneity of rainfall is weak before that of the other
determinants of the water budget and crop yield, typically the lateral and vertical variations of soil
properties and the non-uniformity of irrigation. Conversely, the presence of solar panels may cause
strong spatial heterogeneities possibly compared to that of the water abduction systems used for
irrigation, whose efficiency is estimated from the values of a coefficient of uniformity (Burt et al.,
1997; Playán and Mateos, 2006; Pereira et al., 2002). This paper therefore aims at characterizing the
effective rain distribution in agrivoltaic plots from the calculation of discharge volumes at the outlet
of the panels, depending on their tilting angle. Moreover, the procedure applies to mobile panels
endowed with one degree of freedom, i.e. able to rotate around their support tube according to
predefined strategies, which defines and introduces "dynamic agrivoltaism". Water redistribution in
soils comes in accordance and is briefly described here for coherence checks, it is not the main scope
of the manuscript though crucial for crop growth and irrigation optimisation.

Sect. 2 describes the experimentations conducted on the agrivoltaic plot (Sect. 2.1) and in controlled
conditions (Sect. 2.2), also presenting the AVrain model that predicts rain redistribution by the solar
panels (Sect. 2.3). Sect. 3 shows the experimental and modelling results, discussed in Sect. 4. Sect. 5
gathers the conclusions and openings of this work.



## 2. Material and methods

*2.1. Field experiments*

### 2.1.1. Agrivoltaic plot

The agrivoltaic plot (AV) located on the experimental domain of Lavalette (IRSTEA Montpellier: 43.6466 °N ; 3.8715 °E) covers an area of 490 m$^2$, equipped with four rows of quasi-joined agrivoltaic panels (PV) oriented North-South. The rectangular panels are 2 m long and 1 m wide for a total surface coverage of 152 m$^2$. They are elevated at 5 m and part of a metallic structure supported by pillars separated by 6.4 m, forming square arrays, so as to allow agricultural engines in the agrivoltaic plot. This coverage corresponds to a "half-density" in comparison with a classical free-standing plant. The tilting angle of the PV may vary between -50° and +50° with reference to the flat, horizontal case. This 1-degree of freedom rotation around the horizontal, transverse axis of the panels is ensured by jacks. These may be controlled for solar tracking during daytime or to obey other user-defined time-variable controls. The measurement campaign spreads from October 18$^{th}$, 2015 to October 24$^{th}$, 2016 thus covers a full year. It encompasses 41 monitored rain events, 12 of which recorded with a 1-minute time step, among which 11 exhibit complete and reliable sets of data linked to the incoming and redistributed rain amount, and to the tilting angle of the panels.

### 2.1.2. Effective rain and soil water content measurements

The monitoring of rain amounts in the AV plot is ensured by a series of 21 collectors of 0.3 m diameter, aligned and joined so as to form a continuous line, centered under a PV row, and transverse to it (Fig. 1). In the following, the collectors are termed P01 to P21 from West to East. In addition P0 indicates the rain amount collected in control conditions, just beside the AV plot. All rain amounts collected are expressed as water depths (with 1 mm = 1 L m$^{-2}$). The recordings were made for various angular positions of the PV, either held flat or in abutment ($\pm$ 50°) or during time-variable "avoidance strategies" that mainly consist in minimizing rain interception by the panels by deciding their titling angle from wind direction. Rain amounts in the nearby control zone are measured with a tipping bucket rain gauge (Young 52203, Campbell Sci.). A windvane anemometer (Young 05103-L, Campbell Sci.) allows recording wind direction and velocity.





[Fig.1 about here]

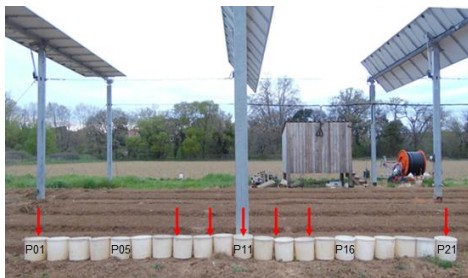



**Figure 1 - Effective rain and soil water content measurement under solar panels. Red arrows indicate the position of neutron probes, on a line parallel to that of the collectors, 1 m before it. Some of the P01 to P21 collectors have been identified on the picture for clarity.**


Soil water content is measured with neutron probes (probe 503DR Hydroprobe, CPN International) until 1 m depth. The soil is predominantly silty and deep. Seven neutron probes were installed at 0.0, 0.5, 1.0 and 3.2 m on both sides of the axis of rotation of the PV row (Fig. 1). Measurements are made once or twice a week on a regular basis but systematically before and after the events.

151

**2.1.3. Experiments in controlled conditions**

A reduced-size agrivoltaic device was built to characterize the influence of the tilting angle of the panels in indoor conditions, monitoring the collected rain amounts in absence of wind with a focus on the lateral redistribution on the width of the panels (Fig. 2). The experimental device consisted of a (2 m x 1 m) panel on a supporting structure of reduced height, allowing tilting angles between 0 and 70°. A rainfall simulator composed of numerous fogging sprays was placed 1.8 m above the flat position of the panel, ensuring quasi-uniform rain conditions on the whole area of the panel, with tested intensities of 20, 35, 60 and 70 mm h$^{-1}$ selected to be representative of the local rain intensities. Water flowing out of the panel was collected on a tilted plane on which 10 half cylinders were fixed, pouring water in the corresponding 10 joined collectors of 0.1 m diameter, covering the width of the panel. The collected amounts were weighted at the end of each test and converted into water depths.





[Fig. 2 about here]

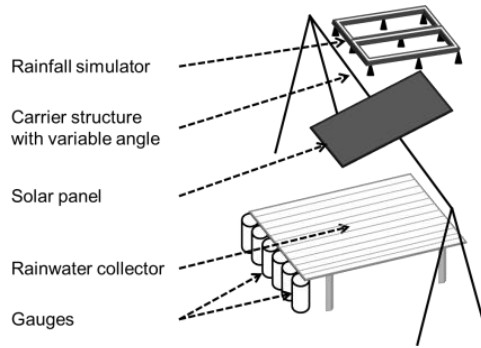


**Figure 2 - Experimental device used for indoor tests, focusing on lateral rain redistribution on the width of the panel, for**
**various combinations of rain intensities and tilting angles of the panel.**

*2.3. Rain redistribution model (AVrain)*
**2.3.1. Model rationale**
The modelling of rain redistribution by solar panels is a geometrical problem describing rain
interception by an impervious surface of length L, tilting angle $\alpha_{PV}$ and height h above the ground, in
which $\alpha_R$ is the angle of incidence of rainfall with respect to the vertical axis and $\theta_R$ denotes the plane
in which the rain falls, with respect to the North in the present case (Fig. 3). The solution is studied in
the vertical (x, z) plane so that the effects in the y direction will be discussed and evaluated but not
explicitly described here. Finally, E is the spacing between the supporting pillars, allowing the
estimation of an equivalent 1-D surface coverage thus the extension of local calculations to the
whole agrivoltaic plot. All notations appear in the Appendix.












[Fig. 3 about here]

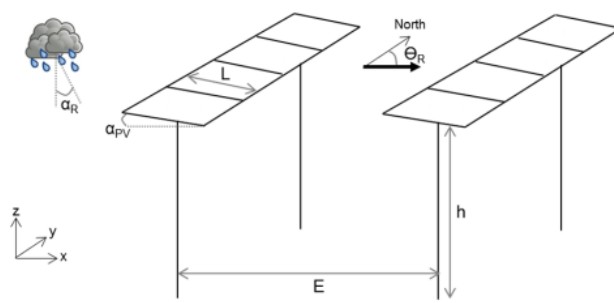



**Figure 3 - Scheme of the simulated scene, indicating the key parameters of the AVrain model that describes rain**
**redistribution by the solar panels on agrivoltaic plots.**

The angle of incidence of rainfall with respect to z may be estimated from the ratio between wind
velocity ($v_w$) and the velocity of the falling rain drops ($v_d$), according to Van Hamme (1992).

$$\tan(\alpha_R) = \frac{v_w}{v_d} \tag{1}$$

In the above, $v_d$ is drawn from the equation proposed by Gunn and Kinzer (1949) for the free-fall limit
velocity of a rain drop in stagnant air, from measurements obtained with the electrical method,
relevant for drop diameters (D) between 0.1 and 5.7 mm:

$$v_d{}^2 = \frac{4}{3} \frac{gD(\rho_s - \rho)}{\rho c} \tag{2}$$

where g is the acceleration of gravity, $\rho_s$ is water density, $\rho$ is air density and c is the drag coefficient.
Drop size distribution has been linked to rain intensity (I) by Best (1950) from previous literature
elements and measurements made by the author:

$$1 - F_{cum} = \exp\left(-\left(\frac{D}{1.3 I^{0.232}}\right)^{2.25}\right) \tag{3}$$

where $F_{cum}$ is the fraction of liquid water in the air comprised in drops with diameters less than D.
The determination of the angle of incidence of rainfall ($\alpha_R$), from given rain intensity (I) and wind
velocity ($v_w$) allows then
- to discriminate the zones impacted by the presence of solar panels from these that will receive the
same rain amounts as in the control zone,





- to calculate the water amount intercepted by the solar panels ($I_{PV}$) in function of I, $\alpha_{PV}$, $\alpha_R$, $\theta_{PV}$ and
$\theta_R$, after Van Hamme (1992):

$$I_{PV} = I \left( \cos \alpha_{PV} - \tan \alpha_R \sin \alpha_{PV} \cos(\theta_{PV} - \theta_R) \right) \tag{4}$$

For simplicity, it is assumed that no significant lateral redistribution occurs on the width of the
panels, resulting in no variation of the outlet flow in the transverse y direction. The relevance of this
hypothesis is justified in the following: the tests in indoor conditions were designed to address this
issue. It is also assumed that the wetting phase of the panels before runoff initiation (somehow the
storage capacity of the panels) has no noticeable effects on the calculations. From observations, for
low tilting angles, the $I_{PV}$ value needed to trigger runoff is 0.2 mm at most which is a weak value
compared to the other values involved in the analysis (and lower than the usual precision of rain
gauges).
Runoff velocity (V) is calculated with the Manning-Strickler formula, hypothesizing flow width is
much larger than flow depth, which makes flow depth approximately equal to the hydraulic radius.
Manning's n coefficient is assumed to be 0.01 $s^{1/3}$ $m^{-1}$ after (Te Chow, 1959) because of the very
smooth glass coating of solar panels.
The parabolic trajectory of the drops falling from the panels is calculated in similar ways for any drop
size (i.e., diameter D) and characterized by the abscissa at which the free falling drop touches ground
(x*) and the free fall duration (t*):

$$
\begin{cases}
x^* = a_x \dfrac{t^{*2}}{2} + V \cos \alpha_{PV}\, t^* + x_0 \\[2mm]
a_x = 2 \cdot 10^{-4} \dfrac{v_w^2}{\dfrac{D}{2}} \\[2mm]
t^* = \dfrac{V \sin \alpha_{PV} + \sqrt{\left(V \sin \alpha_{PV}\right)^2 + 2\, g\, z_0}}{g}
\end{cases}
\tag{5}
$$

where $a_x$ is the acceleration due to wind in the x direction, V is the initial velocity of the fall and $x_0$ is
the abscissa of the edge of the PV.
Drop diameter measurements in control conditions were conducted with a dual-beam
spectropluviometer (Delahaye et al., 2006) and revealed a three-mode distribution of drop diameters

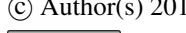



with peaks at D=1.4, 3.8 and 9.3 mm (Fig. 4). However, diameters D > 7.5 mm (Niu et al., 2010) might
be artifacts because rain drops this size would become instable and split in two droplets during their
fall. In the following numerical applications, a fixed diameter of D=1.5 mm is selected as the
reference case for simplicity. However, the sensitivity of the model to D is weak and will be discussed
later.

[Fig. 4 about here]

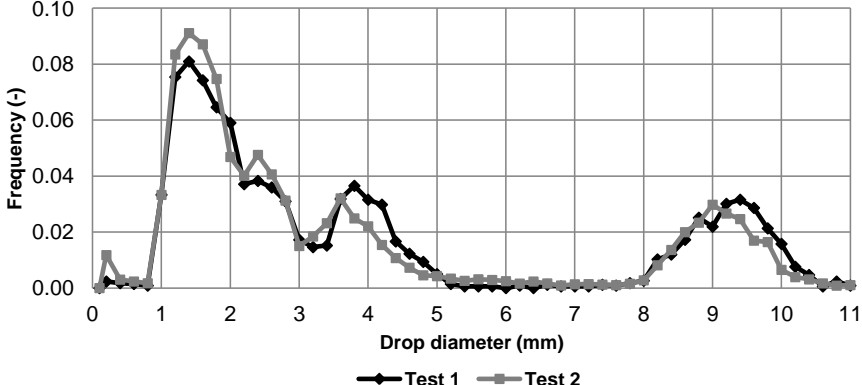



**Figure 4 - Granulometric distribution curve, obtained with a dual-beam spectropluviometer, for the drops falling from the**
**edge of the solar panels. The frequency plotted on the y-axis indicates the count of diameters D observed with respect to**
**the total count (the step is about 0.2 mm in D).**

The AVrain model was developed with the R software to describe 2D (x, z) phenomena in the vertical
plane, hypothesizing negligible effects in the transverse (y) direction (Fig. 1). The time step of AVrain
is 1 minute. The required climatic forcings are: rain intensity (I), wind velocity ($v_w$) and direction ($\theta_R$)
which is assumed identical to rain direction. The input parameters are the geometrical descriptors of
the structure: the height of (the axis of rotation) of the panel (h), its length (L), tilting angle ($\alpha_{PV}$) and
orientation ($\theta_{PV}$), plus the spacing between (pillars supporting the) solar panels (E). Only the tilting
angle can be a function of time as it denotes the control exerted on the system. AV rain allows
calculating rain redistribution (in x) in the form of effective cumulative rainfall amounts in function of
time. A known limitation of this simplified model is that the effects of the secondary slopes of the





panels are not explicitly accounted for, although properly identified by the experiments in controlled
conditions. These have shown that the combination of low tilting angles (i.e. primary slopes $\alpha_{PV}<5°$)
and low rain intensities lead to lateral homogeneities on the edge of the panels, at the risk of
concentrating water fluxes on the lower corner of the panel in extreme cases. However, the
magnitude of this rain redistribution remains limited in the present experimental and is discussed in
the following.

**2.3.2. Sensitivity analysis**
The implementation of solar panels is very likely to affect crop management and irrigation strategies
in the equipped plots, especially because of rain redistribution by the panels. The associated patterns
of spatial heterogeneity may be described by the coefficient of variation (Cv) closely related to the
coefficient that describes the uniformity of water distribution by the irrigation systems (ASAE, 1996;
Burt et al., 1997), thus allowing easy comparisons. The choice of Cv as the target variable for
sensitivity analysis acknowledges spatial heterogeneity is the key descriptor of the effects of solar
panels on rain redistribution on the cultivated plots. In the following, Cv is calculated from the
effective rain amounts (i.e., the cumulative water depths) simulated in the 21 joined collectors along
the x axis. High Cv values indicate strong heterogeneities and Table 1, adapted from ASAE (1996),
recalls the range of Cv values used to qualify the uniformity of water distribution by the irrigation
systems.











Table 1 - Reference values for the coefficient of uniformity of water distribution by irrigation systems, after ASAE (1996)
and Burt et al. (1997). The original values are expressed here as values of the coefficient of variation used to measure the
spatial heterogeneity of rain redistribution by the solar panels.

| Performance | Cv |
|---|---|
| Excellent | < 0.1 |
| Good | 0.1-0.2 |
| Fair | 0.2-0.3 |
| Poor | 0.3-0.4 |
| Unacceptable | > 0.4 |



Using Cv as an indicator allows accounting for two sources of spatial heterogeneity: rain
redistribution by the solar panels (with eventual local effective rain amounts that exceed the
"natural" rain amounts measured in the control zone) and the sheltering effect of solar panels (with
effective rain amounts far lower right under the panels than in the control zone). More into details,
Cv encompasses in a single indicator the spatial heterogeneity observed within the region located
right under a solar panel, i.e. centered on the transverse y axis that connects two supporting pillars,
as clearly seen in Fig. 1 where the P11 is the central collector. The width of the equipped region is E,
selected as the parameter that describes the spacing between panels and further used to estimate
the 1-D spatial coverage of the plot by the panels, also taking place in the sensitivity analysis of the
model.

The Morris (1991) method is used with Cv as the target variable, to estimate the sensitivity of the
AVrain model to assess the effect of its seven main parameters (see Table 2) on the spatial
heterogeneity of rain redistribution by the solar panels. The combined "one-at-a-time" screenings of
the parameter space introduced by Campolongo et al. (2007) have been used to cover a wide set of
possible agrivoltaic installations, keeping all parameters within acceptable, realistic ranges of values.
The "sensitivity" package of R (Pujol et al., 2017) was used to generate the associated 4000
parameter sets, obtained from p=7 parameters with d=500 draws each, dispatched within r=8 levels.





The control parameter (tilting angle $\theta_{PV}$ of the panels) was taken between -70° and +70° but held
fixed for the tested event (P=3.6 mm, $v_w$=0.78 m s$^{-1}$, $\theta_w$=285°, described later).

**Table 2 - Parameters and ranges of values used in the sensitivity analysis of the AVrain model**

| Parameter | Description | Reference | Range | Unit |
|---|---|---|---|---|
| D | Size of the drops falling from the solar panels | 1.5 | 0.1 - 7 | mm |
| E | Spacing between solar panels | 6.40 | 4 - 10 | m |
| FactorP | Multiplying factor for precipitations | 1 | 0.1 - 10 | - |
| FactorV | Multiplying factor for wind velocity | 1 | 0.1 - 10 | - |
| H | Height of the solar panels | 5.00 | 3 - 7 | m |
| L | Lenght of the solar panels | 2.00 | 1 - 3 | m |
| $\theta_{PV}$ | Tilting angle of the solar panels | 0 | -70 - 70 | ° |



*2.4. Control simulations of soil moisture field by Hydrus-2D*

Hydrus-2D (Simunek et al., 1999) may be used to simulate water redistribution in soils for different
fixed tilting angles of the solar panel or strategies in operating the panels. The simulation domain
finds itself in a vertical (x, z) plane, it is centered on the supporting pillar of a panel and covers a total
width of 6.4 m, corresponding to the distance between two consecutive pillars. Hydrus-2D is rather
used here for coherence checks and to gain an overview of water redistribution in soil than for
detailed numerical simulations of the wetting front movements in space and time, thus allowing
simplifying hypotheses on soil structure. The investigated soil depth is 1-m deep, well-known from
numerous local experiment and predominantly silty. It is assumed homogeneous in absence of
significant contrast with depth and presented in Table 3.





**Table 3 - Soil parameters at the Lavalette experimental station used in Hydrus-2D, after Barakat et al. (2017, submitted).**
$\theta_r$ **and** $\theta_s$ **denote respectively the residual and saturated volumetric soil water contents,** $\alpha$ **and** $n$ **are empirical shape**
**parameters of Van Genuchten-Mualem model,** $K_s$ **is the soil hydraulic conductivity at saturation and** $l$ **is a pore**
**connectivity parameter.**

| Depth (cm) | Clay (%) | Silt (%) | Sand (%) | $\theta_r$ (-) | $\theta_s$ (-) | A (cm$^{-1}$) | n (-) | $K_s$ (cm hr$^{-1}$) | l (-) |
|---|---|---|---|---|---|---|---|---|---|
| 0 – 100 | 18 | 42 | 40 | 0.01 | 0.36 | 0.013 | 1.2 | 2.30 | 0.5 |



The AVrain model provides the time-variable forcing data at the soil-atmosphere interface for
Hydrus-2D, divided into five categories and accounting for time-variable tilting angles of the solar
panel (Fig. 5):
- atmospheric conditions for zones not impacted by the presence of the solar panel,
- flux 1 (F1) conditions for zones impacted by the panel and located right under it,
- flux 2 (F2) conditions for zones impacted by the panel but not located under it,
- flux 3 (F3) conditions for zones located under the edge of the panel thus exposed to the largest
effective rain amounts,
- flux 4 (F4) conditions for zones adjacent to these of the F3 conditions but on the sheltered side.

Hydrus-2D currently allows five types of time-variable upper boundary conditions, which suggests
using F2 on both sides of the panel, as indicated in Fig. 5 where only the leftmost position of F2
corresponds to the choices listed above. However, the rightmost position of F2 seems the most
suitable default choice given the known soil filling dynamics and the expected effective rain amounts.
Zero-flux boundary conditions apply on the vertical limits of the domain and free drainage is relevant
for a bottom boundary condition because the water table is several meters under the limit of the
domain. For simplicity, the initial soil water content will be assumed homogeneous, selecting a value
close to the available observations ($\theta$=0.15).






[Figure 5 about here]

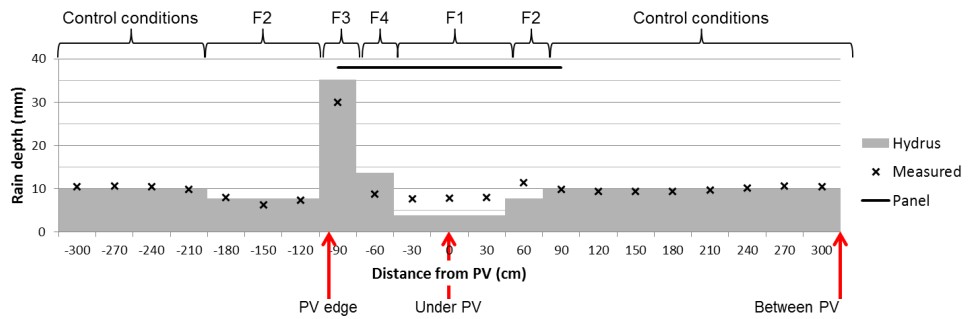


**Figure 5 - Time-variable upper boundary conditions used in Hydrus-2D for the tested rain event, during which the tilting**
**angle of the panels was varied to minimize rain interception (avoidance strategy).**

**3. Results**
*3.1. Rain redistribution measurements on the dynamic agrivoltaic plot*

The influence of variable-tilting angle solar panels on rain redistribution was measured for a wide
series of rain events covering a full year, taking the coefficient of variation (Cv) as the target variable
thus assuming this measure of spatial heterogeneity is the crucial hydrological descriptor in
agrivoltaic contexts. Table 4 gathers Cv values obtained for the most documented rain events in the
available records. It enables comparisons between Cv and the tilting angle (or operating strategy) of
the solar panels, for various rain intensities. The least heterogeneous rain redistributions were
observed for panels in abutment (Fig. 6a, b) mainly due to decreased surface coverage, from 30% for
flat panels to 20% for panels in abutment, resulting in a lesser rain interception. However, the
relevancy of this strategy depends on the angle of the wind with respect to the panels ($\alpha_R$ vs. $\theta_R$)
identifying these as second-order but non-negligible factors, according to which Cv may become
twice as large for panels "facing the wind" or "back to the wind". By contrast, the most
heterogeneous rain redistribution was observed for a flat panel ($\alpha_{PV}=0$) maximizing rain interception
and concentration by the panel (Fig. 6c), collecting 11 times more rain than in the control zone, in the
F4 domain of Fig. 5, with Cv=2.13.



Strategies involving time-variable tilting angles $\alpha_{PV}$ offer multiple possibilities, among which the
previously mentioned "avoidance strategy" is relevant to decrease the spatial heterogeneity (Fig. 6d)
and results in Cv=0.22, that is a fairly good homogeneity according to Table 1. For all the events listed
in Table 4, only the avoidance strategy was able to provide an acceptable level of uniformity in the
agrivoltaic plot, i.e. a spatial heterogeneity than would not need to be corrected on purpose, with a
dedicated precision irrigation device, to ensure equivalent water availability conditions during crop
growth. In all cases, the effective rain depth was more important on the sides of the panel (collectors
9 and 13 in Fig. 1 and Fig. 6). There are non-impacted zones in the free space between panels, where
the effective rain is the same as in the control zone. On the contrary, the sheltering effect is strong
right under the panels and the effective rain is always far lower than in natural conditions.

**Table 4 - Rain events with their identification (ID), date, rain amounts on the control zone (P0), tilting angle of the solar**
**panels ($\alpha_{PV}$) and the associated measured coefficient of variation (Cv) whose highest values indicate the strongest spatial**
**heterogeneities in rain redistribution by the solar panels. In the comments Sect., "avoidance strategy" indicates a time-**
**variable $\alpha_{PV}$ angle to minimize rain interception by the panels in real time.**

| ID | Date | P0 (mm) | $\alpha_{PV}$ | Cv (-) | Comments |
|---|---|---|---|---|---|
| #01 | 18/10/2015 | 4.8 | -50 to 0° | 1.14 | Solar tracking |
| #02 | 07/12/2015 | 5.1 | -50 à -30° | 0.98 | Solar tracking |
| #03 | 12/02/2016 | 14.6 | -50° | 0.97 | Transverse wind (south) |
| #04 | 09/03/2016 | 5.1 | -50° | 0.96 | Facing the wind |
| #05 | 17/03/2016 | 4.1 | +50° | 0.40 | Back to the wind |
| #06 | 21/04/2016 | 3.6 | 0° | 2.13 | Flat panel |
| #07 | 30/04/2016 | 3.0 | 0° | 1.15 | Flat panel |
| #08 | 22/05/2016 | 8.4 | 0° | 0.72 | Flat panel |
| #09 | 28/05/2016 | 13.5 | 0° | 1.28 | Flat panel |
| #10 | 31/05/2016 | 4.5 | 0° | 1.63 | Flat panel |
| #11 | 14/09/2016 | 14.8 | -50 to +50° | 0.22 | Avoidance strategy |
| #12 | 12/10/2016 | 203.6 | 0 ° | 0.51 | Flat panel |







[Figure 6 about here]

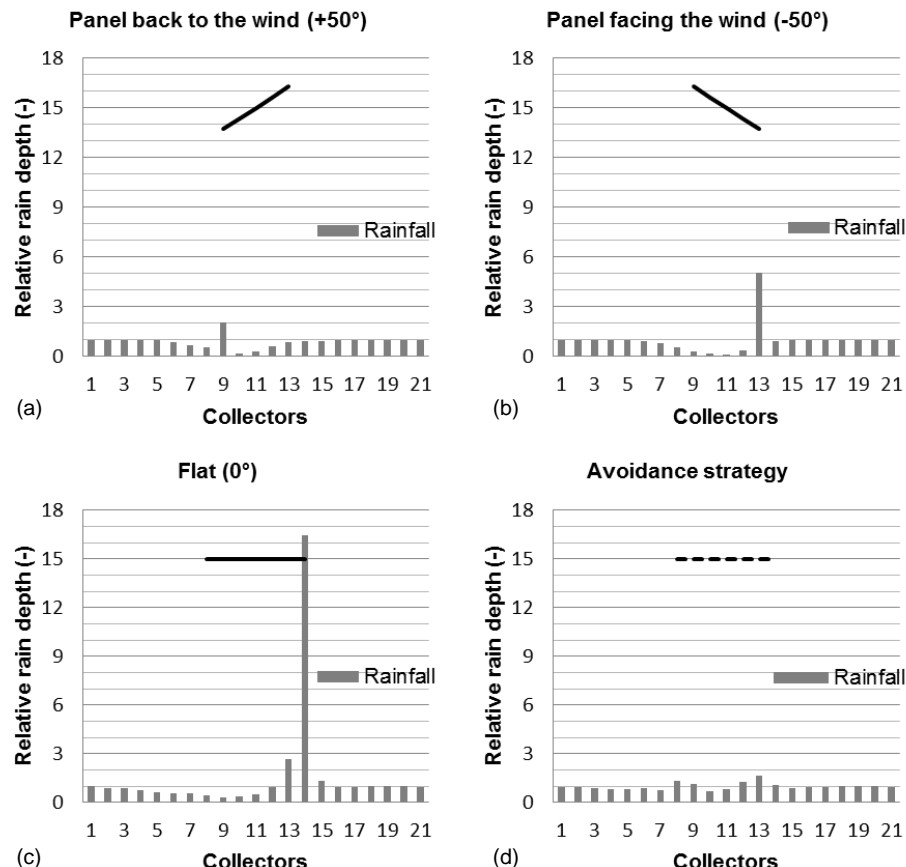

**Figure 6 - Examples of rain redistribution for various rain events, tilting angle and operating strategies of the solar panels,**
**measured in the collectors displayed in Fig. 1. Relative rain depths are given with respect to the control zone where rain**
**amounts are collected in the pluviometer.**

*3.2. Evaluation and sensitivity analysis of the AVrain model*

The rain redistribution model AVrain was tested for 11 rain events involving flat panels, panels in
abutment (either back to the wind or facing the wind) and avoidance strategies, as presented in
Table 5. AVrain describes rain redistribution with a satisfying mean determination coefficient of
$R^2$=0.88. The values of MAPE (Mean Absolute Prediction Error) mostly comprised between 0.1 and

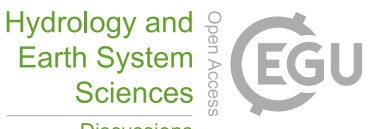

0.3 and regression coefficients greater than 1 indicate that the model tends to overestimate the real
effective rain amounts. However, Fig. 7 shows that the overestimations occur near the drip line (i.e.,
the aplomb) of the panels, totalizing about 25% of the committed errors.

**Table 5 - Performances of the AVrain model that describes rain redistribution by the solar panels, identifying each event**
**(ID), indicating the Mean Absolute Prediction Error (MAPE), Normalized Root Mean Square Error (NRMSE), linear**
**correlation coefficient and coefficient of determination (R²) next to the simulated coeffcients of variation (Cv). The**
**highest Cv values signal the strongest spatial heterogeneities in rain redistribution by the solar panels.**

| ID | MAPE | NRMSE | Linear correlation coefficient | $R^2$ | Cv |
|----|------|-------|-------------------------------|-------|-----|
| #01 | 0.29 | 0.22 | 1.21 | 0.89 | 1.15 |
| #02 | 0.25 | 0.22 | 1.45 | 0.86 | 1.21 |
| #03 | 0.41 | 0.10 | 0.82 | 0.83 | 0.75 |
| #05 | 0.07 | 0.13 | 1.10 | 0.86 | 0.46 |
| #06 | 0.14 | 0.13 | 1.06 | 1.00 | 2.28 |
| #07 | 0.21 | 0.20 | 0.89 | 0.98 | 1.25 |
| #08 | 0.13 | 0.11 | 0.88 | 0.99 | 0.72 |
| #09 | 0.23 | 0.12 | 1.38 | 0.97 | 1.50 |
| #10 | 0.22 | 0.17 | 1.04 | 0.96 | 2.34 |
| #11 | 0.11 | 0.08 | 1.00 | 0.75 | 0.19 |
| #12 | 0.17 | 0.03 | 1.13 | 0.56 | 0.78 |









[Figure 7 about here]

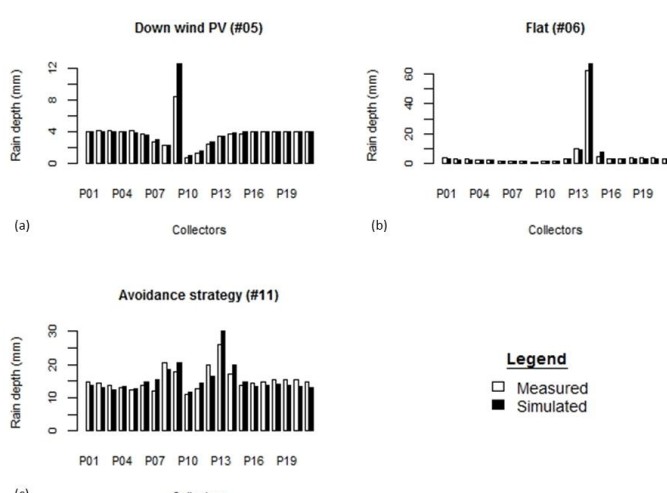


**Figure 7 - Examples of rain redistribution by the solar panels simulated by the AVrain model and compared to field**
**measurements, for three very different events and managements of the solar panels (see Tables 4 and 5 for details).**

The sensitivity analysis of AVrain was conducted with the Morris (1991) method, modified and
improved by Campolong et al. (2007), selecting Cv as the target variable. Figure 8 shows its results,
where $\mu^*$ on the x-axis is the mean of the individual elementary effects (thus the sensitivity of the
parameter tested alone) and $\sigma$ on the y-axis represents the standard deviation of the elementary
effects (thus the sensitivity of the parameter tested in interaction with other parameters). The Morris
plot allows identifying the parameters that have i) a negligible overall effect, denoted by low values
of both $\mu^*$ and $\sigma$, ii) a linear effect, denoted by high values of $\mu^*$, or iii) non-linear or interactive
effects, denoted by high values of $\sigma$. The sensitivity measures ($\mu^*$, $\sigma$) reported in Fig. 8 for each
parameters have been normalized by the value of the highest sensitivity measure ($\sigma$) for the most
sensitive parameter (FV).









[Figure 8 about here]

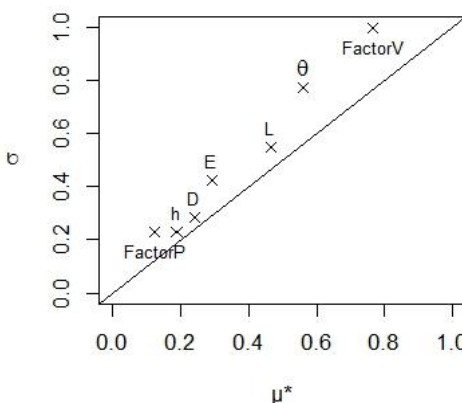


**Figure 8 - Sensitivity analysis of the AVrain model by the Morris (1991) method improved by Campolongo et al. (2007),**
**where µ\* indicates the linear part of the total sensitivity score for each parameter while σ indicates the non-linear or**
**interactive part. In the Morris plot, D is the drop diameter, E the spacing between solar panels, FP the multiplying factor**
**for precipitations with respect to the reference case, FV the multiplying factor for wind velocity with respect to the**
**reference case, h the height of the solar panels, L their length and θ$_{PV}$ their tilting angle (see Table 2 for the reference**
**values and ranges of the parameters). The target variable of the analysis was the coefficient of variation that measures**
**the spatial heterogeneity of rain redistribution by the solar panels. The tested rain event was #06 in Tables 4 and 5.**


The position of the parameters above the 1:1 line in Fig. 8 signals that AVrain is more sensitive to the
interactions between parameters than to individual variations of the parameter values which
reinforces the fact that strong heterogeneities in effective rain amounts most likely occur when
several conditions are met at once, in the forcings (wind direction, drop size), the controls (tilting
angle) and the structure (fixed characteristics of the panels). In particular, the high sensitivity score of
FV compared to the low score of FP indicates that wind velocity tends to influence rain redistribution
patterns far more than rain amounts, likely because wind velocity intervenes in the calculation of the
angle of incidence of rainfall and in that of the trajectory of the drops falling from the panels. The
drop size itself was found of non-negligible but of rather weak influence, although a wide range (0.1
to 7.0 mm) of values was tested. The fact that AVrain is more sensitive to the tilting angle (control
exerted on the system) than to the structure parameters (fixed once selected during the installation)





is a crucial result of the analysis, indicating there is room for optimisation. Conversely, the higher
sensitivity of AVrain to wind velocity than to the tilting angle confirms that the optimisation
strategies should be decided from wind characteristics that dictate the angle of incidence of rainfall.
In an overview of Fig. 8, the Morris method unveils the hierarchy of effects. This proves especially
useful when investigating the interactions between the structure parameters. For example, the
combinations between panels length and spacing (defining surface coverage) are expected to have
more effect on the target variable than the combinations involving panel height, making height a
second-order parameter, at least for the tested (realistic) ranges of values and the chosen target
variable. This conclusion would have been impossible to reach when separately testing the effects of
variations in length, spacing and height of the panels, as proven by Fig. 9 which only acknowledges
adverse effects (on Cv) of length and spacing on the one side, and of height on the other side.

[Figure 9 about here]

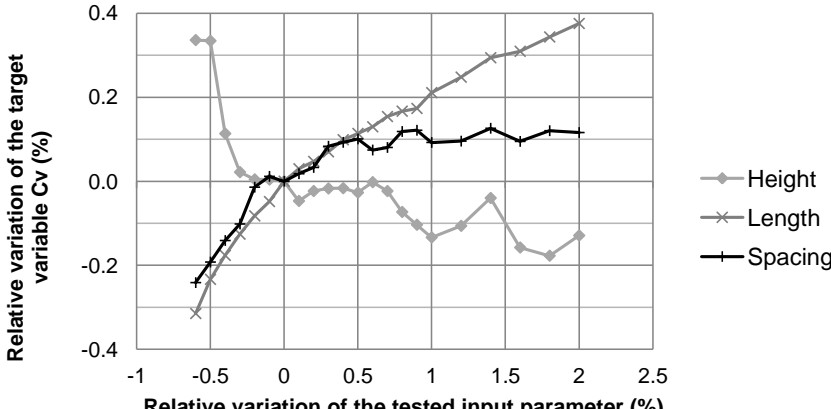


**Figure 9 - Spider diagram showing the influence of the structure parameters (spacing E, height h, length L) of the**
**agrivoltaic installation on the spatial heterogeneity of rain redistribution by the solar panels, from the simulated values**
**of the coefficient of variation (Cv).**

From Fig. 8, the influence of the tilting angle may be expected larger than that of the structure
parameters, anticipating thus that the avoidance strategy (i.e., operating the panels so as to
minimize rain interception) will be prone to significantly reduce Cv whatever the structure



parameters. This point is further investigated by Fig. 10, comparing a flat panel with a piloting of the
panel according to the avoidance strategy, for various combinations of panels length and spacing
(previously proven to have more influence on Cv than the height of the panels). Small-sized panels
with a weak spacing between them is advocated as the best configuration to reduce Cv in avoidance
strategies, simulated to be far more efficient than panel held flat. However, this analysis indicates the
direction to follow when only rain redistribution issues are tackled but external constraints will surely
exist when deciding the in-situ implementation of such agrivoltaic installations, for example in the
form of limit values for the spacing between panels (to allow agricultural activities).

[Figure 10 about here]

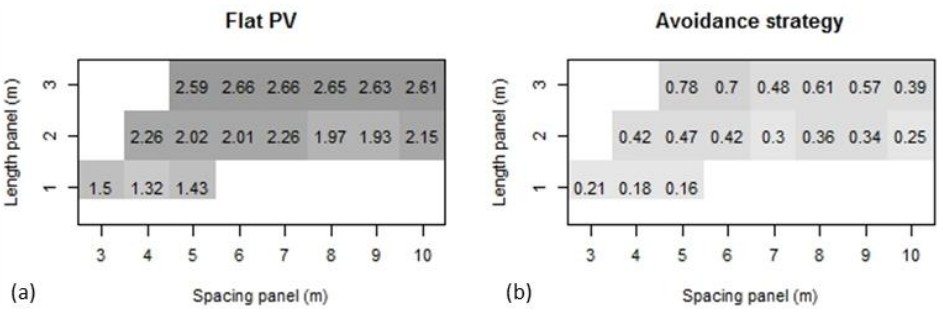

**Figure 10 - Influence of the structure parameters (spacing E, length L) of the panels on the spatial heterogeneity of rain**
**redistribution, from the simulated values of the coefficient of variation (Cv) for panels held flat (a) or operated according**
**to the avoidance strategy (b). The combinations of E and L values may be assimilated to equivalent 1-D surface coverage**
**between 20 and 60% by dividing L by E. Only the realistic combinations have been simulated here: blank cells indicate**
**those that are not.**

*3.3. Rain redistribution in soils*

Water content profiles were measured in the agrivoltaic plot immediately before one of the rain
events, then 6 to 12 hours after it, to identify the dynamics and magnitude of rain redistribution in
soils, as a consequence of rain redistribution on the soil surface. As expected, the spatial
heterogeneity observed on the soil surface is transferred but becomes a bit fuzzy in the first 30 cm of
soil, due to "lateral homogenization" (ponding with significant surface runoff, lateral diffusion



associated with soil dispersivity). But still the spatial patterns are clearly visible within soils, especially
for the flat panels case (Fig. 11a) for which three distinct zones may be identified, i) between panels,
with similar behavior as in the control zone, ii) under panels, with a noticeable sheltering effect thus
drier soils and iii) under the edge of the panels, where the increased soil water content is attributable
to the large effective amounts poured on the soil surface. In Fig. 11a, The maximal soil water storage
variation as observed under the edge of the panels, estimated at 6.7 mm in accordance with the
location of the effective rain amount poured on the soil surface (24.0 mm). Between panels, the
storage variation was 2.0 mm for 3.0 mm of effective rain. Under panels, the storage variation was
4.7 mm for only 1.3 mm of effective rain, which reinforces the hypothesis of lateral redistribution,
either within the soil or at its surface, from the nearby zones. In Fig. 11b, the avoidance strategy
tested for a rain event of 60 mm in the control zone resulted in a maximal storage variation of 91 mm
between panels due to a dryer initial soil water content, 76 mm under panels and 43 mm near the
aplomb of the edge of the panels, while significant ponding was observed.


















[Figure 11 about here]

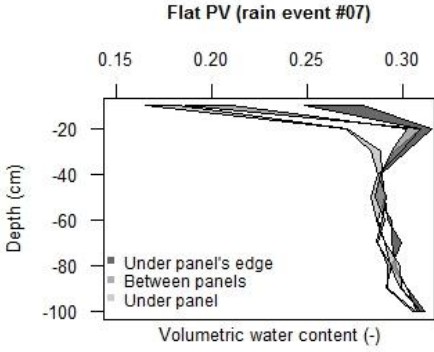

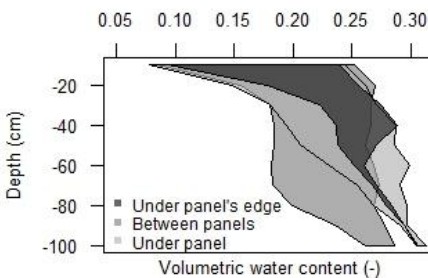


**Figure 11 - Variations of soil water storage in soil regions located near the aplomb of panels edge (dark grey), between panels (medium grey) and under panels (light grey) for different strategies in operating the panels, holding panels flat during rain event #07 (a) or operating them according to the avoidance strategy that minimizes rain interception, during rain event #11 (b). For each case, the leftmost and rightmost line indicate the water content profile before and after the event, respectively. Event #11 was considered as the sum of two successive events for a total rainfall of 60 mm in the control zone.**


The simulation of rain redistribution in soils was made by Hydrus-2D for a single rain event (#11) to
compare the soil water content fields obtained in the flat panel case (Fig. 12a) or when using the
avoidance strategy (Fig. 12b). The time-variable atmospheric conditions required by Hydrus-2D were
provided by the outputs of AVrain at the minute time step, with the five-zone discretization
discussed in Sect. 2.4 and shown in Fig. 5. Starting from a rather dry, realistic and approximately
homogeneous soil water content of θ=0.15, the objective of these exploratory simulations were not
to capture the finest spatial patterns of the wetting front; it was rather to assess if the observed



noticeable differences in rain redistribution trends could easily be reproduced and quantified by
Hydrus-2D. As expected, the flat panel case leads to the creation of a sharp contrast of soil water
content, near the aplomb of the edge of the panel, in the form of a wet bulb that propagates
downward by gravity and sideward by diffusion. This result in the vertical plane is in coherence with a
well-known 3D effect of irrigation, that the vertical and horizontal deformations of the ellipsoidal
bulb will depend on soil properties: coarse soils will produce very elongated bulbs in the vertical
direction while silty soils are likely to produce more significant lateral redistribution. However, the
simulated spatial heterogeneities in soil water content remain very pronounced for the flat panel
case in comparison with the avoidance strategy (Fig. 12b). In this manuscript, the choice of the
coefficient of variation (Cv) to qualify the spatial heterogeneities allowed the reconnection to the
coefficient of uniformity classically used in irrigation science, addressing water delivery on the soil
surface, typically by sprinkler irrigation. Here, Fig. 12a resembles the 2D or 3D patterns characteristic
of surface or subsurface drip irrigation while Fig.12b recalls the quasi-1D patterns of (high-
performance) sprinkler irrigation.

[Figure 12 about here]

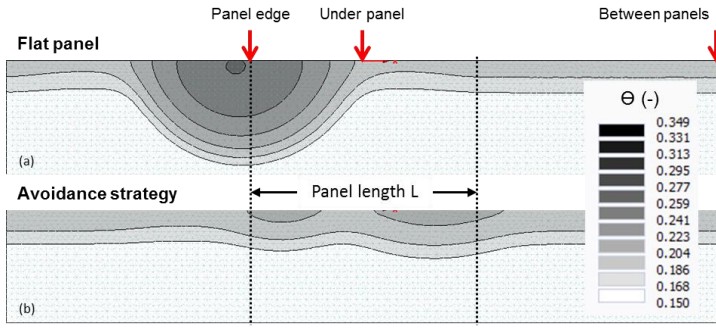

**Figure 12 - Simulation of soil water patterns with Hydrus-2D, in regions located near the aplomb of panels edge, under**
**panels or between panels, when holding the panels flat (a) or operating them according to the avoidance strategy (b) to**
**reduce the heterogeneity of rain redistribution by the panels, during Event #11 (see Tables 4 and 5). The vertical arrows**
**recall the positions of the neutron probes used to collect water content data plotted in Fig. 11.**




*3.4. Effects of the transverse slope of the panels*

The underlying hypotheses made in the construction of the AVrain model led to the formulation of a 2D (x, z) model, discarding thus all phenomena arising from variations in the transverse (y) direction or, at least, not representing them in explicit manner. If relevant, indirect assessments of their effects should still be made, outside AVrain but to investigate if the model stays valid -or in which conditions significant uncertainties may exist on its predictions. Among transverse effects likely to exist in real conditions, only the effects of transverse slopes of the panels were anticipated, observed and deemed significant, though limited to particular contexts. These contexts are summed up in the cases when the tilting angle (i.e. the prevalent slope) of the panels is very low, so that the transverse, secondary slope becomes of the same order.

Tests in controlled conditions were conducted during 15 minutes, under a rain intensity of 20 mm h$^{-1}$. Rain redistribution on the width of the panel appears for tilting angles lower than 20° and the width of the outlet becomes very narrow for tilting angles lower than 5° (Fig. 13). In the latter case, about 90% of the collected water drops from the panel through a 20-cm wide outlet. In the general case, such effects may be explicitly calculated from the slopes (prevalent, secondary) and water depth on the panel.  Such effects are prone to increase the effective rain amounts observed in the field, at the aplomb of the edge of the flat panels (Fig. 6c).





[Figure 13 about here]

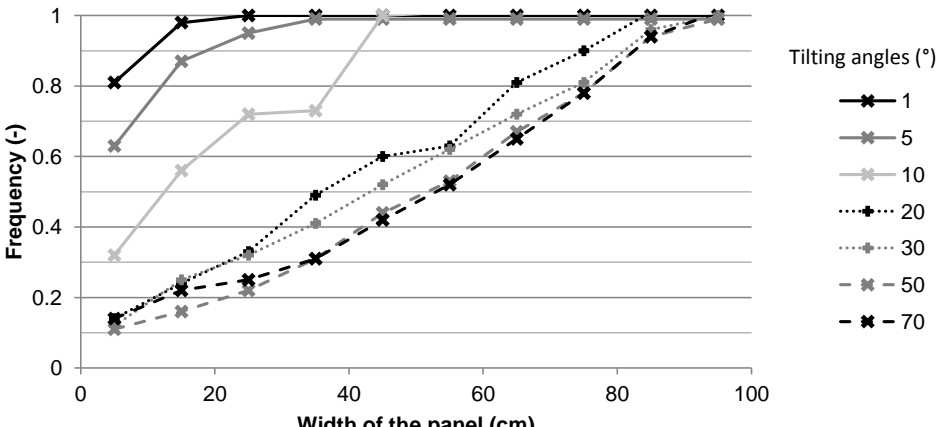


**Figure 13 - Influence of the transverse slope of the solar panels on the lateral rain redistribution on the width of the**

**panel, tested for a 20 mm h$^{-1}$ rain intensity and "prevalent" tilting angles of the panels between 1 and 70°. The results**

**are expressed in cumulative distribution of the collected amounts, at the outlets placed along the width of the panel.**


**4. Discussion**
*4.1. Rain redistribution by the solar panels*

The 2D AVrain model was developed to describe rain interception and redistribution by the solar
panels and fulfills its objectives well: it allows the identification of the sheltered zones and of the
zones in which the effective rain amounts exceed the natural rain amounts of the control zone, with
a correct quantification of the associated fluxes. The angle of incidence of rainfall was found a key
variable in the determination of the spatial patterns of heterogeneity in the effective rain amounts
falling on the ground. This angle is difficult to measure but the equations derived by Gunn and Kinzer
(1949) and Best (1950) allow to estimate it in indirect ways.

If relevant, the AVrain model may be adapted to account for additional geometrical characteristics of
the solar panels, for example to better describe the effects of the secondary (transverse) slope when
it becomes of the same order as the tilting angle of the panels (i.e. their prevalent slope). This is the
typical case in which the secondary slope is prone to increase the heterogeneity of rain redistribution



by redistributing the collected water along the width of the panels. The presence and effect of a
ridge on the length and/or width of the panels could be explicitly modeled with the techniques used
in hydrology for thin flows over a weir. Even if the presence of a small ridge may affect the threshold
of (approximately) 2 mm water depth thought to trigger runoff on the panels (in controlled
conditions and without a ridge), it is hypothesized here that any explicit modelling would not provide
a significant added value, for two reasons: the stored volumetric amounts are weak when the panels
are held nearly flat in absence of rain and the avoidance strategy is recommended when rain occurs.

*4.2. Rain redistribution in soils*

Hydrus-2D was used to simulate rain redistribution in soils, using the spatially distributed output
variables of the AVrain model to provide the required time-variable atmospheric conditions. Five
such conditions at most can be used as climatic forcings for Hydrus-2D, which seemed a limitation for
the present purpose but could be handled, thus with the a posteriori indication that the chosen
"trick" has the value of a good practice. In coherence with the field observations, the simulated fields
of soil water content emphasized the interest of using the avoidance strategy to decrease the spatial
heterogeneities of soil water content in the agrivoltaic plots, confirming thus that the tilting angle of
the panels is a strong control parameter.

Even if the spatial heterogeneity of rain redistribution is less drastic in soils than on the soil surface,
due to lateral diffusion, it remains strong enough to necessitate a dedicated remediation in the form
of precision irrigation, unless the avoidance strategy is used. In other words the avoidance strategy
(that consists in minimizing rain interception and redistribution by commanding the appropriate
time-variable tilting angle of the panels) has implications in the relevant irrigation strategy, making it
less complex. This is an opening to a more global optimisation problem in dealing with the various
sources of heterogeneity, certainly to be compared with the observed heterogeneities in crop yield
on the agrivoltaic plots. Besides the heterogeneities in the forcings (irrigation and rain redistribution)
the modeller will surely have to also address these in soils, for example by means of geophysical





methods that offer the possibility of similar spatial resolutions (e.g., electrical resistivity tomography,
refraction seismology)

*4.3. Rain and crop-induced operation of solar panels*

Some aspects specific to cultivated plots need to be mentioned here, although the primary scope of
this paper is to focus on the hydrological side. The panels left with a low tilting angle (high surface
coverage and rain interception) are prone to have unwanted direct effects on the soil and plants
underneath. For example, leafy vegetables might be damaged by the repeated drop impacts or even
more by the occasional curtains of water falling from the panels a few meters above, even if their
storage capacity is limited. Such problems will typically occur in the morning, when panels are first
operated, being that they are generally left flat during nighttime. They could also occur during heavy
rains, even when using the avoidance strategy, which results in a damped but non-zero flux
concentration near the aplomb of the edges of the solar panels. In the bare soil periods, it is rather
the erosion risk that should be handled, especially "splash erosion" (Nearing and Bradford, 1985;
Josserand and Zaleski, 2003; Planchon and Mouche, 2010) where drop impacts are responsible for
particle detachment and the creation of microtopography, which, in turns, creates pathways for
runoff and further soil degradation processes. Nevertheless, avoidance strategies fed by real-time
wind and precipitation data (collected at a 30 s time step) are powerful means to handle these
issues, certainly to be included in the more general optimisation strategies suitable for the cultivated
agrivoltaic plots.




## 5. Conclusion

Agrivoltaism represents a modern, relevant solution to the growing food and energy demands, associated with a global population increase, especially in the current climate change context. But still there are unresolved issues specific to the implementation of solar panels on the cultivated plots, for example regarding the adaptation of the plants to the forced intermittent shading conditions, or the impact of the panels on the hydrological budget and behavior of the plot. This paper has tackled the pending question of rain redistribution by "dynamic" solar panels, i.e. panels endowed with one degree of freedom in rotating around their supporting axis, so that their tilting angle may vary in time and be controlled on purpose, on a very short term of a few minutes.

A dramatic difference was observed and simulated, in terms of spatial patterns of rain redistribution on the ground, between the case of panels held flat and panels moved according to so-called "avoidance strategies" that consist in minimizing rain interception by the panels during the course of rain events (and eventually adapting the command of the panels to short-term changes in wind and rain conditions within a single event). The avoidance strategies resulted in far lesser coefficients of variation (i.e. heterogeneity measures) used to describe the spatial variations of the effective rain amounts falling on the ground, under the panels, between panels, or near the aplomb of the edges of the panels. The measures of heterogeneity obtained for avoidance strategies had low enough values to be compared with the fairly good uniformity scores used to quantify the ability of irrigation systems to deliver similar water amounts in the different zones of a given plot. Hence, it is likely that the most relevant irrigation strategies will suppress or attenuate the need for precision irrigation within the equipped plots. On the contrary, basic strategies that consist in holding the panels flat induce very strong spatial heterogeneities, with local effective rain amounts that exceed these of the control zone and may be responsible for increased runoff and erosion risks on bare soils, not to mention the risks associated with direct, repeated impacts on the plants that find themselves near the aplomb of the edge of the panels. The flat panel case has one additional disadvantage: the panels are never strictly flat, so that any transverse slope of comparable order will have the consequence of redirecting all the collected water towards a narrow outlet on the width of the panels.

However, the mechanistic AVrain model derived in this paper shows that the control exerted on the tilting angle of the panels is strong enough for the user to cope with most meteorological conditions



(rain intensity, wind direction and velocity) and realistic structure characteristics (height, length and
spacing of the panels) to achieve the targeted short-term event-based optimisation of rain
redistribution. It is very likely that more general and complex methods should be used when
considering both the hydrological budget, crop growth and energy production, as well as seasonal
objectives. To prepare ground, the soil part of the problem has also been investigated here, showing
with Hydrus-2D simulations that rain redistribution patterns in soils resembled these observed on the
soil surface, though less contrasted due to lateral diffusion processes on the soil surface (ponding) or
within soils (at least where significant lateral dispersion coexists with gravity).  Future research leads
include a finer parameterization of Hydrus-2D for a stronger coupling with the results of the AVrain
model, as a verification tool for the adaptation of simpler 1D approaches to model water budget,
irrigation strategies and crop growth in agrivoltaic conditions (Khaledian et al., 2009; Mailhol et al.,
2011; Cheviron et al., 2016) within global optimisation strategies.



**Code availability, data availability, sample availability**

Data collection and model development were performed in the frame of the Sun'Agri2B project that links the Sun'R SAS society with Irstea and other academic or non-academic partners. The copyright on all experimental and theoretical results presented here is governed by the consortium agreement of the Sun'Agri2B project.

**Appendices and supplementary links**

None

**Team list**

The first author is a PhD student, member of both the Sun'R SAS society and the "OPTIMISTE" research team of Irstea Montpellier, France, to which all co-authors also belong. OPTIMISTE stands for Optimization of the Piloting and Technologies of Irrigation, Minimization of InputS, Transfers in the Environment and is one of the research teams in the "G-Eau" joint research unit that addresses water management, actors and usages.

**Author contribution**

Yassin Elamri performed most of the experiments and developed the model, under the supervision of Bruno Cheviron and Gilles Belaud. Annabelle Mange contributed to the first stages of experiments and model development while Cyril Dejean and François Liron helped handling the metrological and technical parts of the work.

**Competing interests**

No known competing interests based on scientific grounds. However, there may be conflicts of interest on commercial grounds with societies other than Sun'R SAS also engaged in agrivoltaic activities.

**Disclaimer**

None

**Special issue statement**

None




**Acknowledgements**


This study was conducted within the frame of the SunAgri2b project, supported by Provence-Alpes-
Côte d'Azur and Rhône-Alpes Regions, CAPI, BPI France, Communauté de Communes Pays d'Aix,
Grand Lyon, the Agence Nationale pour la Recherche et la Technologie. The experimental platform
was co-funded by Irstea, Region Ile-de-France and Paris Entreprises.



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

Yuan, C., G. Gao, and B. Fu. 2017. « Comparisons of Stemflow and Its Bio-/Abiotic Influential Factors
between Two Xerophytic Shrub Species ». *Hydrology and Earth System Sciences* 21 (3):
1421‑38.