# Peer review of "Rain concentration and sheltering effect of solar panels on cultivated plots"

_Hydrology and Earth System Sciences, 2017_

## Referee Comment (RC1) · Anonymous Referee #1 · 24 Aug 2017

General comments: The manuscript describes a study on the rain redistribution by solar panels on soil surface and into the soil profile and the application of a new model to simulate the effective rain amounts on the plot from some forcing data. Also the rain redistribution into the soil using Hydrus-2D model is analysed. The results provide interesting advises on the strategies to adopt for a more appropriate management of solar panels above cultivated plots to obtain an adequate crop production. The references are adequate and up-to-date. The manuscript is well structured and clear. However, I have Specific comments below for which clarification is advised. Also Specific corrections are reported. Once these items are addressed, I believe this article could be suitable for publication.

Specific comments: Line 118: What do you mean with "agricultural engines"? "agricul-

tural machinery" and/or "agricultural equipment"?

Line 129 and Fig. 1: For a better evaluation of distribution uniformity under the solar panels it would be better to use a grid of collectors or replicate the series of 21 collectors more times along the PV row. The results here reported should be considered partial. See also the comment below on the use of coefficient Cv.

Line 159: You affirm that the tested rainfall intensities are representative of the local data; it would be interesting to know the return periods of them.

Line 179: Appendix?

Lines 194 – 225: Please specify the units of measurement of each parameter.

Lines 259-268: I think that the coefficient of variation Cv here cited by the authors is the manufacturer's coefficient of variation or a measure of discharge of a random sample of emitters useful for microirrigation system design. It doesn't describe the uniformity of water distribution by the irrigation system. In this case it is more appropriate the use of the low-quarter distribution uniformity DU also reported in Burt et al. (1997). But in this case it would be better to have data collected in a grid of collectors. The authors are invited to better explain and justify their choice.

Table 1: As regards the Cv see also the comment above. The values here reported are not supported by a proper data analysis.

Line 349: Figure 5 (Please change. It is not Figure 1) is not clear. Is it related to a calibration phase? If it describes the results of the application of the avoidance strategy, why the measured and simulated values in F3 are so high?

Results: See the comment on Cv. The authors should revise the parts in which the Cv is cited according to their revision/choice.

Lines 694-695: The authors must add also the effects of repeated impacts, especially in bare soils, on the soil aggregates with an increase on soil compaction and soil crust

formation

Specific corrections: Line 134: "in abutment"; maybe it is better "inclined, oblique, ..." Line 219: Please substitute with "0.01 s m-1/3 after Chow (1959)" Fig. 4: Please substitute "granulometric" with "drop-size" Lines 251-253: Not clear. Please check the sentence. Line 254: "present experimental". Please check the sentence. Table 3: Please add the units of measurement to $\theta$. Lines 321: A or $\alpha$? Lines 355-358: Not clear. Check the sentence. Table 4: -50 à -30°. Please check. Line 417: Figure 7 not 2. Line 437: Figure 8 not 3. Line 471: Figure 9 not 4. Line 489: Figure 10 not 5. Line 507: Maybe "variation was observed". Line 533: Figure 11 not 6. Line 602: Figure 13 not 7. Line 789: The reference must be moved according alphabetic order. Line 861: The correct citation is "Chow V.T."

---

## Referee Comment (RC2) · Anonymous Referee #2 · 2 Oct 2017

Review: Rain concentration and sheltering effect of solar panels on cultivated plots

Solar PV installations are rapidly increasing globally due to technological advances and policy changes. Colocating solar infrastructure and crops would provide several benefits and should be explored during the planning and construction of large solar installations. In particular, agrivoltaics, if implemented properly, can maximize the efficiency water and land use. For colocating crops in solar installation we need to understand the impact of large installations on local soil-hydrological processes. However, studies investigating these processes (both field and modeling) are limited. Hence this study is significant and timely.

Here the authors developed a model to describe rain interception and redistribution by solar panels to identify sheltered zones and the zones where effective rainfall exceeds

the natural rain amounts. The angle of rainfall was found to be a key factor determining the spatial patterns of heterogeneity of rain water reaching the soil surface. I am not familiar with type of models used in the manuscript, however they sound reasonable and appropriate for the research question. This study is novel and will help in the implementation of best management strategies for optimum water availability for crop plants under solar. he writing could be improved. Some sections are not clear (For example Lines 93-95) and there are some grammatical errors. I found some sections (e.g. results) and some figures (Figure 6) hard to follow. The quality of figures could be improved.

I have some comments:

1. It will be interesting to see how the heterogeneities develop with random positioning of solar panels during rain events. I would imagine leaving the panels in random positions during rain storms should decrease the spatial heterogeneity of water distribution. If the panels stop moving when the rain starts, they will be at different positions during different rainfall events. This is something worth considering (and more energy efficient).

2. Is there energy expenditure/cost for the avoidance strategies described in the manuscript? It will be great if the authors could discuss about this. Are there any other cheaper water redistribution strategies? I feel like retrofitting the panels with some sort of water harvesting structures to redistribute rain might be a cheaper than installing tracking pv panels. Most of the existing solar installations are fixed ones.

3. How does the avoidance strategies affect the dust management /or cleaning of PV panels. The dust accumulation on solar panels is an key factor affecting power output and often the periodic rains are very effective in keeping the panels clean. This is something to consider along with managing the rain-water intercepted by panels.

4. Discuss other factors like shading by panels, that may be more important for crop production than spatial heterogeneity of water distribution in crop fields. Shading cannot be controlled as the panel need to face the sun, while water availability could be managed easily by providing additional irrigation. It would be great if the authors could discuss more on the relative role of these two factors. In arid and semi-arid regions, the redistribution of water could be an important factor compared to shading by panels. In fact, in extremely arid regions the crops might benefit from shading.

5. Do the avoidance strategies or controlling the panels to optimize water distribution impact evapotranspiration from the cropped area?

6. Is there a degree of error that is caused by the diameter of the tipping bucket? I feel that 30 cm width for one point of measurement might cause too big of a mesh when trying to characterize the rainfall distribution of a panel that is 1 to several meters wide at most.

7. For Table 3 and Figure 11, it might be helpful to include the porosity of the soil so that the volumetric water content can be viewed in context of relative saturation of the soil.

8. The word "weak" is used too often where the words "low" or "small" may be more appropriate.

9. It will be great if the authors could discuss the applicability of this study to other locations, in particular in dryland regions where most of the large solar installations are sited. Further, most of the existing solar installations are fixed ones.

---

## Author Comment (AC1) · 17 Oct 2017

We appreciated the careful reading of the manuscript and the constructive comments. We answer the comments below, and we will include the corresponding corrections in the revised version.

R1: Specific comments: Line 118: What do you mean with "agricultural engines"? "agricultural machinery" and/or "agricultural equipment"?

This will be replaced by "agricultural machinery"

R1: Line 129 and Fig. 1: For a better evaluation of distribution uniformity under the solar panels it would be better to use a grid of collectors or replicate the series of 21 collectors more times along the PV row. The results here reported should be consid-

ered partial. See also the comment below on the use of coefficient Cv.

We used a second series of rain collectors indeed for event #12. The results we very close to each others in the two series of collectors (see below).

See as attached document: Figure 1 : rain depths (mm) collected in the two lines of collectors ("Ligne 1, Ligne 2) See as attached document: Figure 2 : correlation between rain depths collected in the two lines

Due to the time necessary to collect all the samples, we could not increase the sampling, so we agree to indicate them as partial. The results could be used nonetheless to check the validity of the model AVrain, where no calibration was necessary to fit on the data (for any of the 12 rain events described in the manuscript).

R1: Line 159: You affirm that the tested rainfall intensities are representative of the local data; it would be interesting to know the return periods of them.

We agree about the relevance of this information, which was missing. The following return periods (based on Montpellier airport station statistics) were obtained: 20mm/h :1 year 35mm/h :3 years 60mm/h :16 years 70mm/h :32 years This information will be added in the manuscript.

R1: Line 179: Appendix?

Sorry, the appendix was present in a previous version. We will remove it in the final version.

R1: Lines 194 – 225: Please specify the units of measurement of each parameter.

The units will be added in the final version.

R1: Lines 259-268: I think that the coefficient of variation Cv here cited by the authors is the manufacturer's coefficient of variation or a measure of discharge of a random sample of emitters useful for microirrigation system design. It doesn't describe the uniformity of water distribution by the irrigation system. In this case it is more appropriate

the use of the low-quarter distribution uniformity DU also reported in Burt et al. (1997). But in this case it would be better to have data collected in a grid of collectors. The authors are invited to better explain and justify their choice.

There are indeed different manners to estimate the uniformity of application. Notably Christiansen's uniformity coefficient defined as: UC=100%(1-(average deviation from the averagedepth)/(overall average depth of application)) is a widely used method of uniformity evaluation for sprinkler systems (Burt et al. 1997). Our coefficient of variation is merely related to UC: Cv=1-UC The advantage of using UC (or Cv, which is equivalent) is that the uniformity of rain application due to panels can be compared to the one caused by usual spray irrigation systems. It is however interesting to use other uniformity indicators as suggested by the reviewer, such as DU. DU will use the average of the lower quartile, so the rain depths obtained under the solar panels. Obviously, DU and UC (or Cv) are correlated, as also reported in Burt et al. (1997). This can be observed in the attached figure (figure 3). Therefore, both methods will give similar conclusions. The final version will include a remark about that.

R1: Table 1: As regards the Cv see also the comment above. The values here reported are not supported by a proper data analysis.

We agree that these values are not reported in a unique document. The article of Burt et al. (1997) indicates the value of 80% for fair uniformity. It also indicates the linear behavior between DU and CV (equations 14, 15): DU≈1-K Cv with K≈1. Therefore we selected the threshold of Cv= 0.2 for fair distribution. This is supported by other studies using Christiansen coefficient: "Catch can measurements are used to determine the uniformity of a sprinkler irrigation system. Christiansen (1942) developed a numerical index representing the system uniformity of overlapping sprinklers. This uniformity coefficient (UC) is a percentage on a scale of 0 to 100 (absolute uniformity). A uniformity coefficient of 80 is considered by many to be the minimum acceptable performance. Higher uniformity coefficients are usually needed with intensively maintained ornamentals. Catch can measurements are also used to illustrate water distribution or

patterns." (http://horticulture.oregonstate.edu/system/files/onn110110.pdf)

The values indicated in the manuscript were taken from ASAE standards (see http://edis.ifas.ufl.edu/ae094 - see table in Figure 4).

Actually, the most interesting information is the threshold to characterize an acceptable uniformity. A value of Cv =0.2 for this threshold is consistent for all papers cited in the manuscript.

R1: Line 349: Figure 5 (Please change. It is not Figure 1) is not clear. Is it related to a calibrwas ation phase? If it describes the results of the application of the avoidance strategy, why the measured and simulated values in F3 are so high?

The figure number has been changed. The data refer to both measurements and simulations, which are consistent. For this event, the wind intensity was very high so the rain orientation (N-S direction) made it impossible to achieve the rain avoidance strategy since the panels can only rotate along the N-S axis. This causes the heterogeneity to remain high whatever the panel orientation strategy.

R1: Results: See the comment on Cv. The authors should revise the parts in which the Cv is cited according to their revision/choice.

The part related to Cv will be changed as suggested by reviewer.

R1: Lines 694-695: The authors must add also the effects of repeated impacts, especially in bare soils, on the soil aggregates with an increase on soil compaction and soil crust formation

These impacts may appear indeed. This will be added as suggested.

R1: Specific corrections: Line 134: "in abutment"; maybe it is better "inclined, oblique, : : :"

This will be changed to "inclined"

[Figure]

R1: Line 219: Please substitute with "0.01 s m-1/3 after Chow (1959)"

This will be corrected as suggested

R1: Fig. 4: Please substitute "granulometric" with "drop-size"

This will be corrected as suggested

R1: Lines 251-253: Not clear. Please check the sentence. Line 254: "present experimental". Please check the sentence.

These two sentences will be rewritten as follows: "These have shown that the combination of low tilting angles (i.e. primary slopes ïĄąPV<5°) and low rain intensities lead to lateral homogeneities on the edge of the panels. In these cases, this leads to concentrate water fluxes on the lower corner of the panel. However, the impact on the water balance (and its heterogeneity) is limited due to the low magnitude of the corresponding rainfall amounts. is discussed in section 4.1."

R1: Table 3: Please add the units of measurement to Ïť.

The unit (m3/m3] will be added

R1: Lines 321: A or _?

A will be removed

R1: Lines 355-358: Not clear. Check the sentence.

This sentence will be clarified as follows: "The influence of variable-tilting angle solar panels on rain redistribution was measured thanks to a wide series of rain events covering a full year. For each event, we put a focus on the spatial heterogeneity, which is assumed to be a crucial issue for the hydrological balance of solar panels on crops. This heterogeneity is characterized with the coefficient of variation Cv of rain depths."

R1: Table 4: -50 à -30_. Please check.

This will be corrected

R1: Line 417: Figure 7 not 2.

This will be corrected

R1: Line 437: Figure 8 not 3.

This will be corrected

R1: Line 471: Figure 9 not 4.

This will be corrected

R1: Line 489: Figure 10 not 5.

This will be corrected

R1: Line 507: Maybe "variation was observed".

This will be corrected

R1: Line 533: Figure 11 not 6.

This will be corrected

R1: Line 602: Figure 13 not 7.

This will be corrected

R1: Line 789: The reference must be moved according alphabetic order.

This will be corrected

R1: Line 861:The correct citation is "Chow V.T."

This will be corrected

[Figure]

Fig. 1. rain depths (mm) collected in the two lines of collectors ("Ligne 1, Ligne 2)

[Figure]

**Fig. 2.** correlation between rain depths collected in the two lines

**DU vs Cv**

$R^2 = 0.7293$

**Fig. 3.** Correlation between DU and Cv

Table 1. Microirrigation system uniformity classifications based on emitter discharge rates[1].

| Classification | Uniformity, $U_s$ (%) |
|---|---|
| Excellent | above 90% |
| Good | 90%–80% |
| Fair | 80%–70% |
| Poor | 70%–60% |
| Unacceptable | below 60% |

[1]Adopted from ASAE (1996a).

**Fig. 4.** Table from ASAE standards (reference values for uniformity)

---

## Author Response (AR1)

Montpellier, October 18th, 2017

**For the attention of Lixin Wang, editor, HESS**

**Subject**: revised version submission
**By**: Yassin Elamri, Bruno Cheviron, Annabelle Mange, Cyril Dejean, François Liron, Gilles Belaud

Dear Lixin Wang,

We are very pleased to re-submit the attached manuscript "Rain concentration and sheltering effect of solar panels on cultivated plots» as a research paper to be published in *Hydrology and Earth System Sciences*.
This manuscript was revised according to the comments of reviewers. We think we replied all the comments, and included the required corrections.
Notably:
- We rewrote some unclear paragraphs, as pointed out by reviewers;
- We clarified the justification about the uniformity indicator;
- We added suggestions of reviewers (regarding return periods of rain, solar position strategy, soil structure evolution…);
- Edition was improved (grammar and expressions were corrected).

Please find further the point-by-point responses to the comments.
We hope this manuscript will fulfil all your requirements for its publication, and look forward to reading from you.

Yours sincerely,

Bruno Cheviron and Gilles Belaud
Corresponding authors
* * *
UMR G-eau –361 rue JF Breton, BP 5095, 34196 Montpellier cedex 5, FRANCE

**Anonymous Referee #1**
General comments: The manuscript describes a study on the rain redistribution by solar panels on soil surface and into the soil profile and the application of a new model to simulate the effective rain amounts on the plot from some forcing data. Also the rain redistribution into the soil using Hydrus-2D model is analysed. The results provide interesting advises on the strategies to adopt for a more appropriate management of solar panels above cultivated plots to obtain an adequate crop production. The references are adequate and up-to-date. The manuscript is well structured and clear.

However, I have Specific comments below for which clarification is advised. Also Specific corrections are reported. Once these items are addressed, I believe this article could be suitable for publication.

Authors: We appreciated the careful reading of the manuscript and the constructive comments. We answer the comments below, and will include the corresponding corrections in the revised version.

Specific comments: Line 118: What do you mean with "agricultural engines"? "agricultural machinery" and/or "agricultural equipment"?
Authors: This has been replaced by "agricultural machinery"

Line 129 and Fig. 1: For a better evaluation of distribution uniformity under the solar panels it would be better to use a grid of collectors or replicate the series of 21 collectors more times along the PV row. The results here reported should be considered partial. See also the comment below on the use of coefficient Cv.
Authors: We used a second series of rain collectors indeed for event #12. The results we very close to each others in the two series of collectors (see below).

[Figure]

Figure 1 : rain depths (mm) collected in the two lines of collectors ("Ligne 1, Ligne 2)

UMR G-eau –361 rue JF Breton, BP 5095, 34196 Montpellier cedex 5, FRANCE

[Figure]

**Figure 2 : correlation between rain depths collected in the two lines**

Due to the time necessary to collect all the samples, we could not increase the sampling, so we agree to indicate them as partial. The results could be used nonetheless to check the validity of the model AVrain, where no calibration was necessary to fit on the data (for any of the 12 rain events described in the manuscript).

Line 159: You affirm that the tested rainfall intensities are representative of the local data; it would be interesting to know the return periods of them.

Authors: We agree about the relevance of this information, which was missing. The following return periods (based on Montpellier airport station statistics) were obtained:
20mm/h ⇔1 year
35mm/h ⇔3 years
60mm/h ⇔16 years
70mm/h ⇔32 years
This information has been added in the manuscript.

Line 179: Appendix?
Authors: Sorry, the appendix was present in a previous version. We will remove it in the final version.

Lines 194 – 225: Please specify the units of measurement of each parameter.
Authors: The units have been added in the final version.

Lines 259-268: I think that the coefficient of variation Cv here cited by the authors is the manufacturer's coefficient of variation or a measure of discharge of a random sample of emitters useful for microirrigation system design. It doesn't describe the uniformity of water distribution by the irrigation system. In this case it is more appropriate the use of the low-quarter distribution uniformity DU also reported in Burt et al. (1997). But in this case it would be better to have data collected in a grid of collectors. The authors are invited to better explain and justify their choice.

UMR G-eau –361 rue JF Breton, BP 5095, 34196 Montpellier cedex 5, FRANCE

Authors: There are indeed different manners to estimate the uniformity of application. Notably Christiansen's uniformity coefficient defined as:

$$UC = 100\% \left(1 - \frac{average\ deviation\ from\ the\ average\ depth}{overall\ average\ depth\ of\ application}\right)$$

is a widely used method of uniformity evaluation for sprinkler systems (Burt et al. 1997). Our coefficient of variation is merely related to UC: Cv=1-UC

The advantage of using UC (or Cv, which is equivalent) is that the uniformity of rain application due to panels can be compared to the one caused by usual spray irrigation systems.

It is however interesting to use other uniformity indicators as suggested by the reviewer, such as DU. DU will use the average of the lower quartile, so the rain depths obtained under the solar panels. Obviously, DU and UC (or Cv) are correlated, as also reported in Burt et al. (1997). This can be observed in the figure below.

[Figure]

Therefore, both methods will give similar conclusions. The final version will include a remark about that.

Table 1: As regards the Cv see also the comment above. The values here reported are not supported by a proper data analysis.

Authors: We agree that these values are not reported in a unique document. The article of Burt et al. (1997) indicates the value of 80% for fair uniformity. It also indicates the linear behavior between DU and CV (equations 14, 15): $DU \approx 1 - K\ Cv$ with K≈1. Therefore we selected the threshold of Cv= 0.2 for fair distribution. This is supported by other studies using Christiansen coefficient:

*"Catch can measurements are used to determine the uniformity of a sprinkler irrigation system. Christiansen (1942) developed a numerical index representing the system uniformity of overlapping sprinklers. This uniformity coefficient (UC) is a percentage on a scale of 0 to 100 (absolute uniformity). A uniformity coefficient of 80 is considered by many to be the minimum acceptable performance. Higher uniformity coefficients are usually needed with intensively maintained ornamentals. Catch can measurements are also used to illustrate water distribution or patterns."* (http://horticulture.oregonstate.edu/system/files/onn110110.pdf)

The values indicated in the manuscript were taken from ASAE standards (see http://edis.ifas.ufl.edu/ae094) :

UMR G-eau –361 rue JF Breton, BP 5095, 34196 Montpellier cedex 5, FRANCE

Table 1. Microirrigation system uniformity classifications based on emitter discharge rates[1].

| Classification | Uniformity, $U_s$ (%) |
|---|---|
| Excellent | above 90% |
| Good | 90%–80% |
| Fair | 80%–70% |
| Poor | 70%–60% |
| Unacceptable | below 60% |

[1]Adopted from ASAE (1996a).

Actually, the most interesting information is the threshold to characterize an acceptable uniformity. A value of Cv =0.2 for this threshold is consistent for all papers cited in the manuscript. We removed the whole table, keeping the value of 0.2 as a unique reference threshold to qualify the importance of heterogeneity.

Line 349: Figure 5 (Please change. It is not Figure 1) is not clear. Is it related to a calibrwas ation phase? If it describes the results of the application of the avoidance strategy, why the measured and simulated values in F3 are so high?
Authors: The figure number has been changed.
The data refer to both measurements and simulations, which are consistent. For this event, the wind intensity was very high so the rain orientation (N-S direction) made it impossible to achieve the rain avoidance strategy since the panels can only rotate along the N-S axis. This causes the heterogeneity to remain high whatever the panel orientation strategy.

Results: See the comment on Cv. The authors should revise the parts in which the Cv is cited according to their revision/choice.
Authors: The part related to Cv has been changed as suggested by reviewer.

Lines 694-695: The authors must add also the effects of repeated impacts, especially in bare soils, on the soil aggregates with an increase on soil compaction and soil crust formation
Authors: These impacts may appear indeed. This has been added as suggested.

Specific corrections:
Line 134: "in abutment"; maybe it is better "inclined, oblique, : : :"
Authors: This has been changed to "inclined"
Line 219: Please substitute with "0.01 s m-1/3 after Chow (1959)"
Authors: This has been corrected as suggested
Fig. 4: Please substitute "granulometric" with "drop-size"
Authors: This has been corrected as suggested

 Lines 251-253: Not clear. Please check the sentence. Line 254: "present experimental". Please check the sentence.
Authors: These two sentences has been rewritten as follows:
"These have shown that the combination of low tilting angles (i.e. primary slopes $\alpha_{PV}$<5°) and low rain intensities lead to lateral homogeneities on the edge of the panels. In these cases, this leads to concentrate water fluxes on the lower corner of the panel. However, the impact on the water

balance (and its heterogeneity) is limited due to the low magnitude of the corresponding rainfall amounts. is discussed in section 4.1."

Table 3: Please add the units of measurement to Ө.
Authors: The unit (m3/m3] has been added
Lines 321: A or _?
Authors: A  has been removed
Lines 355-358: Not clear. Check the sentence.
This sentence has been clarified as follows:
"The influence of variable-tilting angle solar panels on rain redistribution was measured thanks to a wide series of rain events covering a full year. For each event, we put a focus on the spatial heterogeneity, which is assumed to be a crucial issue for the hydrological balance of solar panels on crops. This heterogeneity is characterized with the coefficient of variation Cv of rain depths."

Table 4: -50 à -30_. Please check.
Authors: This has been corrected

Line 417: Figure 7 not 2.
Authors: This has been corrected
Line 437: Figure 8 not 3.
Authors: This has been corrected
Line 471: Figure 9 not 4.
Authors: This has been corrected
Line 489: Figure 10 not 5.
Authors: This has been corrected
Line 507: Maybe "variation was observed".
Authors: This has been corrected
Line 533: Figure 11 not 6.
Authors: This has been corrected
Line 602: Figure 13 not 7.
Authors: This has been corrected
Line 789: The reference must be moved according alphabetic order.
Authors: This has been corrected
Line 861:The correct citation is "Chow V.T."
Authors: This has been corrected.

Anonymous Referee #2

Reviewer2: Rain concentration and sheltering effect of solar panels on cultivated plots Solar PV installations are rapidly increasing globally due to technological advances and policy changes. Colocating solar infrastructure and crops would provide several benefits and should be explored during the planning and construction of large solar installlations. In particular, grivoltaics, if implemented properly, can maximize the efficiency water and land use. For colocating crops in solar installation we need to understand the impact of large installations on local soil-hydrological processes. However, studies investigating these processes (both field and modeling) are limited. Hence this study is significant and timely.

Here the authors developed a model to describe rain interception and redistribution by solar panels to identify sheltered zones and the zones where effective rainfall exceeds the natural rain amounts. The angle of rainfall was found to be a key factor determining the spatial patterns of heterogeneity of rain water reaching the soil surface. I am not familiar with type of models used in the manuscript, however they sound reasonable and appropriate for the research question. This study is novel and will help in the implementation of best management strategies for optimum water availability for crop plants under solar. he writing could be improved. Some sections are not clear (For example Lines 93-95) and there are some grammatical errors. I found some sections (e.g. results) and some figures (Figure 6) hard to follow. The quality of figures could be improved.

Authors: We thank the reviewer for his/her constructive remarks. They have been considered in the revised version.

Reviewer2: I have some comments:

1. It will be interesting to see how the heterogeneities develop with random positioning of solar panels during rain events. I would imagine leaving the panels in random positions during rain storms should decrease the spatial heterogeneity of water distribution. If the panels stop moving when the rain starts, they will be at different positions during different rainfall events. This is something worth considering (and more energy efficient).

Authors: This is an interesting suggestion. This could indeed decrease the heterogeneity at the plot scale and on the long term, e.g. when considering an annual balance. In the case of agrivoltaism, the objective is to limit the impact of panels on crop growth, so it is preferred:
  1- to minimize the heterogeneity for each event, using the avoidance strategy which is also energy efficient
  2- if irrigation is necessary, to adapt the irrigation amount to the actual needs, considering that different depths would be necessary in different zones considering the effect of rain distribution.
In the Mediterranean, it is frequent that rainfall events are spaced by several weeks, so random position may not guarantee global homogeneity during the cropping season.
We will add a remark about this suggestion, which could be relevant in some contexts (more frequent rainfalls, no irrigation).

R2: 2. Is there energy expenditure/cost for the avoidance strategies described in the manuscript? It will be great if the authors could discuss about this. Are there any other cheaper water redistribution strategies? I feel like retrofitting the panels with some sort of water harvesting structures to redistribute rain might be a cheaper than installing tracking pv panels. Most of the existing solar installations are fixed ones.

Authors: The cost of operation is very low compared to the energy produced by panels. Trackers were installed initially to control radiation. This has become quite common for solar farms, considering that the cost of trackers is largely covered by the gain on electricity production. In the case of agrivoltaism, trackers allow controlling radiation received by crops. Therefore, trackers are not only justified by the rain strategy, but the presence of trackers allows implementing advanced control strategies at almost no cost. Rain harvesting is indeed an idea that could reveal to be interesting is some regions. It needs collecting and storing water, which appeared too costly in our climatic context.
A remark will be added on this point.

R2: 3. How does the avoidance strategies affect the dust management /or cleaning of PV panels. The dust accumulation on solar panels is an key factor affecting power output and often the periodic rains are very effective in keeping the panels clean. This is something to consider along with managing the rain-water intercepted by panels.

Authors: This issue was not considered. We think that the avoidance strategy would favor the cleaning of panels: they are only moved after an amount of 0.2mm, then the panels have a maximum inclination of 50°. The threshold of 0.2mm appears quite low and could raised at a value selected by experience.
A remark about this suggestion will be added in the final version.

R2: 4. Discuss other factors like shading by panels, that may be more important for crop production than spatial heterogeneity of water distribution in crop fields. Shading can- not be controlled as the panel need to face the sun, while water availability could be managed easily by providing additional irrigation. It would be great if the authors could discuss more on the relative role of these two factors. In arid and semi-arid regions, the redistribution of water could be an important factor compared to shading by panels. In fact, in extremely arid regions the crops might benefit from shading.
Authors: We fully agree with the reviewer. These factors are analyzed is a separate paper under review (so, not referenced here). Different strategies can be applied during rainfall events compared to the normal strategy (which are electricity- or crop-oriented strategies). We realized that hydrological heterogeneity was high in our Mediterranean context (with characteristics of semi-arid climate), so it was important to characterize this heterogeneity and its link to panel operation, and, in turn, to derive a model predicting rain distribution under movable panels. This led to design the avoidance strategy that applies only during the rain events.
The other points developed by the reviewer need specific instrumentation to understand the effect of radiation variations on crop evapotranspiration, and specific developments to adapt classical water balance models to the conditions imposed by solar panels. AVrain model will be an input of the crop-water balance model, but other adaptations will be needed regarding crop development and water consumption. Obviously, this must be treated in a separate paper.

R2: 5. Do the avoidance strategies or controlling the panels to optimize water distribution impact evapotranspiration from the cropped area?

Authors: The avoidance strategy is followed only during rainfall events. The evaporation rates are low during these periods, since radiation reduces to diffuse radiation. Therefore, evapotranspiration will remain low whatever the control strategy. Once rainfalls have stopped, other strategies are applied (e.g., solar tracking).

R2: 6. Is there a degree of error that is caused by the diameter of the tipping bucket? I feel that 30 cm width for one point of measurement might cause too big of a mesh when trying to characterize the rainfall distribution of a panel that is 1 to several meters wide at most.

Authors: The size of the buckets was selected to characterize the overall heterogeneity of the amount of rain received by the soil. Due to redistribution in the soil (see section 3.3) and to the sensitivity of the phenomenon to inherently dispersed factors (wind velocity and drop size), we considered that it was not meaningful to characterize the heterogeneity at a smaller scale. However, following the reviewer's suggestion, we checked that the conclusions about overall heterogeneity (and the relative importance of the 5 zones as defined in attached (Figure 1) were robust.

[Figure]

Obviously, the maximum rain amount depends on the size of the bucket, but the overall heterogeneity is weakly influenced by this size (here Cv around 2.3 for 30cm, 2.8 for 10cm).

R2: 7. For Table 3 and Figure 11, it might be helpful to include the porosity of the soil so that the volumetric water content can be viewed in context of relative saturation of the soil.

Authors: The porosity will be included in the figure and in the legend of Table 3

R2: 8. The word "weak" is used too often where the words "low" or "small" may be more appropriate.

Authors: We replaced "weak" as suggested by reviewer

R2: 9. It will be great if the authors could discuss the applicability of this study to other locations, in particular in dryland regions where most of the large solar installations are sited. Further, most of the existing solar installations are fixed ones.

Authors: The AVrain model is based on a mechanistic approach, so it is predictive, and fully applicable to any other context, especially in areas with high radiation (and large rain intensities).

UMR G-eau –361 rue JF Breton, BP 5095, 34196 Montpellier cedex 5, FRANCE

The model can be applied to fixed panels too. For such panels, a small width and an inclination of 20° will be recommended in order to avoid excessive intensities at the panel borders.

Of course, the avoidance strategy may be adapted, as discussed above, although we think that, in the case of agrivoltaism (say, solar panels with an objective of crop production), avoidance strategy is the most appropriate in order to avoid undesired crop yield variability.

Note that agrivoltaism is a quite new concept (Dupraz et al. 2011), justified by the search of areas to produce electricity without competing with food production. It raises new questions, but it also brings references that may be useful to more traditional solar farms.